# Tuning of the Anti-Breast Cancer Activity of Betulinic Acid via Its Conversion to Ionic Liquids

**DOI:** 10.3390/pharmaceutics16040496

**Published:** 2024-04-03

**Authors:** Paula Ossowicz-Rupniewska, Joanna Klebeko, Irina Georgieva, Sonia Apostolova, Łukasz Struk, Svetla Todinova, Rumiana Dimitrova Tzoneva, Maya Guncheva

**Affiliations:** 1Department of Chemical Organic Technology and Polymeric Materials, Faculty of Chemical Technology and Engineering, West Pomeranian University of Technology, Piastów Ave. 42, 71-065 Szczecin, Poland; paula.ossowicz@zut.edu.pl (P.O.-R.); joanna.klebeko@gmail.com (J.K.); 2Institute of Biophysics and Biomedical Engineering, Bulgarian Academy of Sciences, Acad. G. Bonchev Str., Bl. 21, 1113 Sofia, Bulgaria; igeorgieva@biomed.bas.bg (I.G.); sonia_apostolova@yahoo.com (S.A.); todinova@abv.bg (S.T.); rtzoneva65@gmail.com (R.D.T.); 3Department of Organic and Physical Chemistry, Faculty of Chemical Technology and Engineering, West Pomeranian University of Technology, Piastów Ave. 42, 71-065 Szczecin, Poland; lukasz.struk@zut.edu.pl; 4Institute of Organic Chemistry with Centre of Phytochemistry, Bulgarian Academy of Sciences, Acad. G. Bonchev Str., Bl. 9, 1113 Sofia, Bulgaria

**Keywords:** betulinic acid, ionic liquids, MCF-7, cytotoxicity, cologenicity, DSC

## Abstract

Betulinic acid (BA) is a natural pentacyclic triterpene with diverse biological activities. However, its low water solubility limits its pharmaceutical application. The conversion of pharmaceutically active molecules into ionic liquids (ILs) is a promising strategy to improve their physicochemical properties, stability, and/or potency. Here, we report the synthesis and characterization of 15 novel ILs containing a cation ethyl ester of a polar, non-polar, or charged amino acid [AAOEt] and an anion BA. Except for [ValOEt][BA], we observed preserved or up to 2-fold enhanced cytotoxicity toward hormone-dependent breast cancer cells MCF-7. The estimated IC50 (72 h) values within the series varied between 4.8 and 25.7 µM. We found that the most cytotoxic IL, [LysOEt][BA]_2_, reduced clonogenic efficiency to 20% compared to that of BA. In addition, we evaluated the effect of a 72 h treatment with BA or [LysOEt][BA]_2_, the most cytotoxic compound, on the thermodynamic behavior of MCF-7 cells. Based on our data, we suggest that the charged amino acid lysine included in the novel ILs provokes cytotoxicity by a mechanism involving alteration in membrane lipid organization, which could be accompanied by modulation of the visco–elastic properties of the cytoplasm.

## 1. Introduction

Carcinogenesis is one of the most complex biological processes. The ability to sustain proliferative signaling, avoid growth suppressors and immune destruction, enable replicative immortality, promote tumor inflammation, activate invasion and metastasis, and induce or accelerate vascularization are some of the general characteristics of cancer cells [1]. Additionally, cancer cells are characterized by genome instability and mutations, dysregulated cell metabolism, and the ability to resist cell death [1]. Breast cancer (BC) is the second leading cause of death in women, and it accounts for 12.5% of all new annual cancer cases worldwide [2,3]. All markers, along with clinicopathological variables, are widely used to predict disease progression and determine outcomes, and in the selection of an appropriate treatment. Due to BC heterogeneity, the first-line treatment of the disease typically involves one or multiple chemotherapeutics having distinct modes of action such as antimetabolites, immunotherapeutics, hormonal/endocrine drugs, DNA alkylating agents, ion modulators, antimitotic agents, etc. [4,5]. The long-term survival of breast cancer patients depends on the tumor subtype, stage at the time of diagnosis, specific tumor markers, and the effectiveness of treatment. Conventional chemotherapy has limitations due to the lack of selectivity, off-target effects, and the development of drug resistance, which can occur even after the first course of treatment [6]. Chemotherapeutics may also cause severe acute or chronic adverse effects like neutropenia, leucopenia, thrombocytopenia, hematological toxicity, cardiotoxicity, and skin toxicity, or cause heart, kidney, or liver failures [4,6]. Therefore, new and effective drug candidates with minimal side effects are always in high demand. Currently, more than 60% of medicines are natural or natural-derived products [7,8]. Plants, marine organisms, bacteria, and fungi are abundant sources of biomolecules with pharmacological potential and have been extensively studied [7,8,9]. Natural products often display good selectivity and target multiple cancer signaling pathways, exhibiting synergistic efficacy when combined with traditional anticancer drugs [10,11]. In addition, natural products are excellent candidates for sensitizing tumor cells to chemotherapy and enhancing the effectiveness of existing drugs [11].

Betulinic acid (BA) is a lupane-type pentacyclic triterpene that is found in the bark of several plants and is industrially isolated from plants of *Betula* species [12]. It has shown great pharmacological potential due to its wide variety of biological activities, including anti-tumor, anti-inflammatory, anti-oxidative, anti-diabetic, anti-malarial, antibacterial, antiviral, antifungal, and some other properties [12,13,14,15]. Among these, the anti-tumor properties of BA have been studied the most extensively. Initially, it was discovered that BA selectively inhibits human melanoma cells amongst other cancer cells [16]. Since then, many studies have reported its effectiveness against various other types of cancer, including leukemia, malignant head and neck cancer, colorectal carcinoma, breast tumor, lung carcinoma, liver cancer, cervical cancer, ovarian cancer, and other carcinomas [12,17,18,19,20]. BA inhibits the tumor growth of various BC cell lines, including MCF-7, MDA-MB-231, MDA-MB-453, BT474, and T47D, via diverse mechanisms [21]. It induces cell cycle arrest by suppressing cyclin and topoisomerase expression, triggers apoptosis through the mitochondrial or intrinsic pathway by caspase-dependent and caspase-independent mechanisms, and inhibits the NF-kB signaling pathway. In vivo BA mediates the regulation of JAK/STAT, VEGF, EGF/EGFR, TRAIL/TRAIL-R, AKT/mTOR, and ubiquitination pathways, or inhibits some of the transcription factors, thereby reducing the expression of matrix proteinase-2 and -9 (MMP2 and MMP9). It stimulates the expression of the tissue inhibitor of metalloproteinases 2 (TIMP2); inhibits angiogenesis; interferes with the mechanisms of resistance, tumor progression and metastasis; reprograms glucose and lipid metabolism; and reduces the level of tumor-induced proinflammatory cytokines, i.e., modulates the tumor microenvironment [21,22]. In addition, BA can affect estrogen receptor (ER) signaling in ER-positive BC cells [23]. It should be noted that BA can inhibit multidrug resistance (MDR) proteins, thereby overcoming MDR, increasing BC cell chemosensitivity to other chemotherapeutics, and reducing their side effects [24,25,26]. Despite the huge pharmacological potential of BA, particularly as an anti-BC chemotherapeutic agent, its medicinal application is still limited due to its poor solubility in water and hence low bioavailability (2.3%), and to its short half-time but long elimination half-life in vivo [27]. Compared to known drugs, BA is classified as a moderate to poorly permeable compound in a model of intestinal mucus [27]. The synthesis of organic salts, specifically ILs containing BA paired with amino acids, represents a strategic approach to overcome the limitations associated with BA’s poor solubility and bioavailability. By harnessing the unique properties of amino acids and optimizing the formulation of BA, we can enhance its therapeutic potential as an anti-breast cancer agent and pave the way for its practical medical application.

In recent years, significant efforts have been dedicated to enhancing the suitability of BA for pharmaceutical and nutraceutical applications. Conjugations of BA with synthetic or bio-polymers, nanoscale formulations, and the use of excipients are some of the strategies that have been explored to enhance the physicochemical properties (solubility and partition property) and chemical stability of BA; to increase its activity, strength, and selectivity; to adjust its pharmacokinetic parameters such as onset and duration of action; to reduce or eliminate its toxicity and adverse reactions; and last but not least to develop novel formulations that can secure intellectual property [28,29,30,31,32].

The conversion of active pharmaceutical ingredients (APIs) into ionic liquid (IL) formulations is a relatively novel and effective approach for drug delivery, tailoring APIs’ solubility and bioavailability, improving their thermal and storage stability, increasing dissolution, resolving problems with their polymorphic conversion, regulating drug release, boosting APIs’ permeability, and/or modulating their cytotoxicity on tumor cells [33,34,35,36].

Ionic liquids consist of large organic cations and organic or inorganic anions with melting temperatures below 100 °C and low vapor pressure. Their physicochemical properties can be tuned by the selection or appropriate modifications of either of the ions.

In recent years, much attention has been drawn to the pharmaceutical and biotechnological application of ILs as a drug delivery system, API-based ILs, ILs as media for biomolecule extractions or media for biocatalytic reactions, and as enhancers of protein stability, activity, and solubility [37,38,39,40,41]. The most studied groups of ILs as stabilizing agents, permeation enhancers, or delivery systems of therapeutic protein and pharmaceutically active compounds, are ILs containing 1, 3-dialkyl imidazolium, cholinium, or alkylammonium cations and various anions [42]. The major concerns over their usage are related to their biocompatibility, cytotoxicity, viscosity, etc. [43]. Hence, ILs containing at least one biocompatible ion are of current interest. In addition, numerous second and third generation ILs have shown excellent anticancer, antimicrobial and/or antiviral activity [44,45,46].

Experimental and modeling studies on the mechanism of action of ILs suggested that some of them can diffuse into the cellular membrane, modify the arrangement of the phospholipids, and affect the membrane potential, fluidity, and viscosity, thereby influencing various biochemical and biophysical processes in cells [47]. Depending on the structure of the ILs, they could also alter the membrane permeability and nucleate pores, or even disrupt the membrane [47]. ILs could interact with or bind to membrane proteins affecting the function of some of the membrane transporters; they could also interfere with RNA and deregulate the synthesis of the cytoplasmatic proteins and/or disrupt or diffuse into the cell nucleus [47].

There are only a few reports on the conversion of BA into ILs and how this affects its physicochemical properties and biological activities. For example, ChaIla et al. reported IC_50_ (72 h) values of choline and benzalkonium betulinates 3- to 4-fold lower than those for BA against melanoma A375 cells, neuroblastoma SH-SY5Y cells, and breast adenocarcinoma MCF-7 cells [48]. Similarly, Williams et al. found that both ILs can inhibit the growth of prostate cancer, but the effect of benzalkonium betulinate is much stronger than that of BA [49]. Silva et al. reported that in comparison to BA, trihexyl(tetradecyl)phosphonium betulinate showed decreased or preserved cytotoxicity but enhanced selectivity when tested against numerous cancer cell lines such as hepatoma Hep G2, osteosarcoma MG63, breast cancer T47 D, lung cancer A459, and colon cancer cells RKO [50].

The available data on the effect of BA-ILs on cancer cells is limited. However, this approach seems promising and research in this direction is worth conducting.

Based on the literature, we assume that esters of AAs could be appropriate cations due to their biocompatibility and overall safety. Recently, Shimul et al. thoroughly investigated the potential of AAE-based phenolic ionic liquids as green harmless solvents of poorly water-soluble bioflavonoids. Screening amongst a large number of tested ILs has shown that the proline ethyl ester ferulate, p-coumarate, vanilate, and 4-hydroxybenzoate facilitate the solubilization of luteolin. They improve the organoleptic properties of luteolin and preserve its antioxidant, antimicrobial, and food preservation properties. Therefore, they have potential in the food industries [51]. On the other hand, Thiele and Lipsky found that methyl esters of Leu or Glu and Phe induce monocyte cytotoxicity in two distinct pathways. This process is more complex for LeuOMe and Glu(OMe)_2_ and involves multistep enzyme reactions. The three AA esters showed different selectivity against NK cells, cytotoxic T lymphocytes, and mononuclear phagocytes. Interestingly, PheOMe causes lyse to myeloid tumor cells such as U937, HL60, and THP-1, while LeuOMe had no effect on these cells [52]. In a model of the intestinal tract based on a colon carcinoma cell monolayer (Caco-2 cells), Kobayashi and Watanabe found that aromatic amino acid ethyl esters can decrease/inhibit the transcellular permeation of proteinous allergens, thus affecting the properties of the epithelial barrier and the immune responses to luminal allergens [53]. Many epithelial cancers are rich in peptide transporters; therefore, in some cases, an AA ester prodrug may significantly facilitate the cellular uptake of the parent drug [54]. In addition, amino acid derivatives of floxuridine and gemcitabine are reported to have enhanced chemical and enzymatic stability, and, consequently, higher toxicity. This is due to the resistance of glycosidic bond cleavage in the molecule. Moreover, compared to the parent molecule, AA ester prodrug formulations showed an eight to eleven times greater permeability across Caco-2 monolayer models [55].

In this manuscript, we describe the synthesis of a series of 15 novel ionic liquids containing a cation ethyl ester of a polar, non-polar, or charged amino acid [AAOEt] and an anion betulinic acid [BA]. The structures were confirmed by ^1^H and ^13^C Nuclear Magnetic Resonance (NMR) and attenuated total reflectance Fourier transform infrared spectroscopy (ATR-FTIR). Specific rotation was used to assess the optical purity of the AAs. The thermal stabilities of [AAOEt][BA] were assessed by thermogravimetric analysis (TG) and phase transition temperatures by differential scanning calorimetry (DSC). The water solubility of the BA-ILs was tested. We evaluated the effect of the modification on BA cytotoxicity against a hormone-dependent breast cancer line MCF-7. In addition, for the most and least cytotoxic [AAOEt][BA], we monitored their ability to suppress MCF-7 colony formation. We assessed the effect of a 72 h treatment with BA or [LysOEt][BA]_2_, the most cytotoxic compound, on the thermodynamic behavior of MCF-7 cells.

## 2. Materials and Methods

### 2.1. Materials

#### 2.1.1. For the IL Preparation

Unless otherwise specified, all reagents and materials used were obtained without additional purification from commercial suppliers. Betulinic acid (BA) (≥98%), 1-octanol, and acetonitrile (for HPLC, gradient grade) were obtained from Sigma-Aldrich (Steinheim am Albuch, Germany). L-aspartic acid, L-isoleucine, L-leucine, L-methionine, L-proline, L-threonine, and L-valine of purity ≥98% were purchased from FluoroChem (Derbyshire, UK). L-serine (≥98.5%), L-phenylalanine (≥98.5%), L-proline (≥98.5%), and L-cysteine (≥98%) were provided by Carl Roth (Karlsruhe, Germany). L-alanine (>99.0%) was acquired from Bachem (Bubendorf, Switzerland) and L-lysine (≥98.0%, anhydrous) from Glentham Life Sciences (Corsham, UK). L-phenylalanine (99%) was provided by Alfa Aesar (Ward Hill, MA, USA). L-tryptophan (>98.5%) was purchased from TCI. Thionyl chloride (99.5%) was provided by Across Organic Geel (Geel, Belgium). Ammonium hydroxide solution 25% (NH_3_.H_2_O) of analytical grade was purchased from StanLab (Lublin, Poland). Ethanol (99.8%) was provided by Merck (Darmstadt, Germany). Methylene chloride and DMSO of high purity were provided by Chempur (Gliwice, Poland). Deuterated dimethyl sulfoxide (DMSO-d_6_) (99.8%) was obtained from Deutero GMBH (Kastellaun, Germany).

#### 2.1.2. Materials for Cell Culturing

Human breast cancer cell line MCF-7 was obtained from the CLS Collection (Cell Lines Service GmbH—Dr.-Eckener-Str. 8, 69,214 Eppelheim, Germany, Catalog number 300273, Lot number 300273-2120). Identity by DNA fingerprint was tested by STR analysis. The non-tumorigenic human mammary epithelial cell line (MCF-10A) was a kind gift from Assoc. Prof. Dr. Neus Colomina and Assoc. Prof. Dr. Eloi Garí from the Cell Cycle Group at the Biomedical Research Institute of Lleida Dr. Pifarré Foundation (IRB Lleida), and the University of Lleida, Spain. Eagle’s minimum essential medium (EMEM) with 2 mM L-glutamine and 1 mM sodium pyruvate, antibiotic mixture containing 100 U/mL Penicillin, 100 µg/mL Streptomycin, 0.25 µg/mL Amphotericin B, Dulbecco’s Modified Eagle Medium/Nutrient Mixture F-12 (DMEM/F-12), hydrocortisone, hEGF, cholera toxin, fetal bovine serum (FBS), and 3-(4,5-Dimethylthiazol-2-yl)-2,5-diphenyltetrazolium bromide (MTT) were purchased from Sigma-Aldrich (Germany). Insulin (Humulin R) was obtained from Lilly Global Headquarters, Eli Lilly and Company (Indianapolis, IN, USA 46285). The MycoStrip^TM^ mycoplasma detection kit was purchased from InvivoGen (Toulouse, France).

### 2.2. Methods

#### 2.2.1. Synthesis of L-Amino Acid Ethyl Ester Betulinates

The procedure for obtaining L-amino acid ethyl ester betulinates ([AAOEt][BA]) was based on a three-step method described previously [56]. The general scheme of the reaction is presented in Figure 1. In the first step, the esterification of the appropriate L-amino acid (using ethyl alcohol) was conducted in the presence of thionyl chloride (AA/SOCl_2_) as a chlorinating agent and reaction catalyst at a molar excess of 1:3 (AA/SOCl_2_). SOCl_2_ was added at a temperature of about 0 °C. The reaction was conducted at 40 °C for 24–48 h. The hydrochloride was isolated from the reaction mixture by distillation of ethanol, chloroform extraction, and diethyl ether rinsing described in previous papers [57,58]. Overnight, the obtained hydrochloride was desiccated at 60 °C under 20 mbar of pressure. In the subsequent step, L-amino acid alkyl ester hydrochloride was neutralized using three molar equivalents of an aqueous solution containing 25% ammonium hydroxide. Diethyl ether was used to extract the formed ester. The solvent was then distilled in a vacuum. In the final stage, the obtained L-amino acid alkyl ester reacted with an equal amount of betulinic acid. The reaction was conducted in methylene chloride for 30 min at room temperature. The solvent was then extracted by distillation at 25 °C with reduced pressure. The obtained BA-ILs were dried in a vacuum oven at 50 °C for 24 h.

#### 2.2.2. Identification and Characterization of [AAOEt][BA]

##### NMR

NMR spectra were captured on a BRUKER DPX-400 spectrometer (Billerica, MA, USA) at 400 and 100 MHz for ^1^H and ^13^C NMR, respectively. DMSO-d_6_ was used as a solvent.

##### FTIR–ATR

The FTIR spectra were recorded using a Nicolet 380 from Thermo Scientific (Waltham, MA, USA) with a diamond crystal. The measurement was conducted between 4000 and 400 cm^−1^ with a resolution of 4 cm^−1^.

##### Elemental Analysis

The elemental composition was determined using a Thermo Scientific^TM^ FLASH 2000 CHNS/O Analyzer (Waltham, MA, USA) mounted with a thermal conductivity detector. The samples were prepared in crucibles made of tin (CHNS analysis) or silver (O analysis) and weighed with an accuracy of 0.000001 g. The composition of specific elements was determined using the calibration curve. L-cysteine, L-methionine, and sulphanilamide were used as standards in CHNS mode, while acetanilide and benzoic acid were used for calibration in O-mode, respectively.

##### TG

Thermal stability was determined using the Netzsch Proteus Thermal Analysis TG 209 F1 Libra apparatus (Selb, Germany). The test was conducted in the 25–1000 °C temperature range at a heating rate of 10 ° min^−1^ under air (flow rate of 25 cm^−3^min^−1^) and nitrogen (flow rate of 10 cm^−3^min^−1^). The weight of the samples was 5–7 mg.

##### DSC of BA-ILs

The phase transition temperatures of the obtained compounds were determined using a TA Instruments DSC 2500 differential scanning calorimeter (New Castle, DE, USA). The measurements were conducted at a temperature between −60 °C and the temperature at which the compound began to decompose, as determined by TG analysis. The 5–8 mg samples were analyzed in closed aluminum crucibles. The heating/cooling/heating rate was 5 °min^−1^.

##### Specific Rotation

The optical rotation was analyzed using an AUTOPOL IV Polarimeter from Rudolph Research Analytical (Hackettstown, NJ, USA) with a 589 nm wavelength for 0.5% (in DMSO) concentrations of compounds at 20.0 ± 0.1 °C. The angular rotation was determined with an accuracy of 0.001°.

##### Estimation of the Water Solubility

The water solubility of betulinic acid and its derivatives was evaluated following the described method [59,60]. For this purpose, 1 mg of the substance was dissolved in 100 μL of DMSO. The stock solution was diluted with 10 μL potions of deionized water until the sediment was observed. 10 μL portions were added to produce dilutions corresponding to 5 times, 10 times, 20 times, 50 times, and 100 times the original concentration. Based on the added portions of solvent, the solubility of the studied compounds was classified as weak, middle, good, very good, or excellent.

#### 2.2.3. Cell Culture

The human breast cancer cell line MCF-7 was cultured in EMEM medium supplemented with 10% FBS, 2 mM L-glutamine, 1 mM sodium pyruvate, 100 U/mL Penicillin, 100 µg/mL Streptomycin, and 0.25 µg/mL Amphotericin B. The non-tumorigenic human mammary epithelial cell line MCF-10A was grown in DMEM/F-12 with 10% FBS, 2 mM L-glutamine, 1 mM sodium pyruvate, 100 U/mL Penicillin, 100 µg/mL Streptomycin, 0.25 µg/mL Amphotericin B, 10 µg/mL insulin, 500 µg/mL hydrocortisone, 20 ng/mL hEGF, and 20 ng/mL cholera toxin. The cells were maintained at 37 °C in a humidified atmosphere with 5% CO_2_ and were routinely tested for mycoplasma using a MycoStrip^TM^ mycoplasma detection kit. In all performed experiments, both cell lines were mycoplasma negative.

#### 2.2.4. MTT Assay

The colorimetric reduction assay MTT was performed to determine the cytotoxic effect of BA and all tested BA-ILs. The assay was conducted as described by Mosmann [61]. Briefly, MCF-7 (or MCF-10A) cells were seeded in 96-well plates at a density of 5 × 10^3^ cells per well. BA and BA-ILs were dissolved in DMSO to obtain stock solutions of 10 mM, which were further diluted with a cell culture medium for preparing working concentrations where the content of DMSO did not exceed 1%. After 24 h incubation, cancer cells were treated with the following concentrations of BA and BA-ILs: 0.5 µM, 1 µM, 2 µM, 3.5 µM, 5 µM, 10 µM, 20 µM, 40 µM, and 60 µM, for 72 h. Non-treated cancer cells were used as a control. After 72 h of treatment, 20 µL of an MTT (with a concentration of 5 mg/mL) was added to each well, and cells were incubated for 3 h in a thermostat at 37 °C. The obtained formazan crystals were dissolved by adding 100 µL/mL of 5% formic acid diluted in isopropanol (≤100%). The absorbance was measured at a wavelength of 570 nm by a plate reader (Tecan Infinite F200 PRO, Mannedorf, Switzerland). Cell viability (%) of MCF-7 (or MCF-10A) cells treated with BA or [AAOEt][BA] was calculated as a percentage of the control using the formula: (OD treated cells/OD control cells) × 100.

The half maximal inhibitory concentration (IC_50_) values were determined by non-linear regression analysis using GraphPad Prism version 5.0 (GraphPad Software, Inc., La Jolla, CA, USA) from three independent experiments.

#### 2.2.5. Colony Formation Assay

Clonogenic assays were performed, and surviving fractions were calculated as described by Franken et al. [62]. Briefly, MCF-7 cells were seeded in 6-well plates at a concentration of 500 cells/well and allowed to adhere for 48 h and subsequently treated with 2.5–12.5 µM concentration of BA and selected BA-ILs (Day 1). The BA and BA-ILs were replenished every 4 days. On Day 10, cells were fixed with 10% formalin in phosphate-buffered saline (PBS), pH 7.4 for 30 min at room temperature (RT), washed with PBS and stained with 0.1% Crystal violet solution in ddH_2_O, 30 min, RT, followed by a final thorough wash in ddH_2_O. The colonies were then imaged and analyzed with the open-source platform Fiji [63].

#### 2.2.6. Cell Morphology Analysis

The changes in cell morphology after a 72 h treatment with BA and [AAOEt][BA] were observed under an inverted microscope (MEIJI, Japan) with objective 10 (Phase DM, MEIJI, Japan) and camera Optikam B1 Digital (Optika, Ponteranica, Bergamo, Italy).

#### 2.2.7. DSC of MCF-7 Cells

DSC measurements were performed on a DASM4 (Biopribor, Pushchino, Russia) in a highly sensitive microcalorimetric system with a cell volume of 0.47 mL. For DSC analysis, exponentially growing MCF-7 cells were harvested and seeded in 100 mm cell culture dish plates at a density of 9 × 10^5^ cells per dish and were left to grow for 24 h at 37 °C in a humidified incubator containing 5% CO_2_. Then, cells were treated with BA or [LysOEt][BA]_2_ at concentrations corresponding to IC_50_ for 72 h, which are 11.5 µM and 4.8 µM respectively. After treatment, cancer cells were harvested by trypsinization and centrifuged together with their cell culture medium. Floating dead cells were also collected. The cells were then thoroughly washed twice with PBS (pH 7.4) and counted using a hemocytometer. Approximately 1 × 10^7^–3 × 10^7^ cells in PBS were used for the calorimetric assay. Samples were transferred to the DSC cells and a constant pressure of 2 atm was applied throughout the experiment to prevent degassing of the suspensions. All experiments were performed at a scanning rate of 1 °C min^−1^ in the range of 30–120 °C. The second heating scan was performed to evaluate the reversibility of the thermal transition. The thermograms were obtained by subtracting the second scans from the first and normalized to the DNA concentration (determined spectrophotometrically using the extinction coefficient of 0.020 (μg/mL)^−1^ cm^−1^ for double-stranded DNA at 260 nm) [64]. The excess heat capacity curves were analyzed using the software package Origin Pro 2018 (OriginLab Corporation, Northampton, MA, USA). The following calorimetric parameters were determined: transition temperature (Tm) and excess heat capacity (cPex) of successive thermal transitions.

## 3. Results and Discussion

### 3.1. Synthesis and Characterization of [AAOEt][BA]

We synthesized 15 ILs comprised of the amino acid ethyl ester cation and anion BA (Figure 1). The BA derivatives were obtained with high yields (85–97%). All the compounds obtained are new and have not been described in the literature. Therefore, they were fully identified (NMR, FTIR, and elemental analysis), and basic physicochemical properties (thermal stability and phase transition temperatures) were determined.

All the compounds obtained were identified using ^1^H and ^13^C NMR, ATR-FTIR, and elemental analysis. The results of NMR spectroscopy and elemental analysis also confirmed the purity of the BA-ILs. Individual results are provided in the Appendix A (ESI). The ESI (Appendix A) contains all ^1^H and ^13^C NMR, FTIR spectra, TG, and DSC curves for non-modified BA and [AAOEt][BA]. For complete identification—NMR, FTIR spectra assignments, and elemental analysis—see the Appendix A.

Figure 2 summarizes the ^1^H NMR spectra of the obtained betulinates. Both betulinic acid and amino acid ethyl ester signals are seen. Due to poor solubility in other deuterated solvents, the spectra were made in DMSO-d_6_, which, unfortunately, made the signals from the amino group invisible. The confirmation of obtainment of an ionic structure was the presence of signals from the NH3+ group in the spectrum, which was visible in the [ValOEt][BA] spectrum (see Appendix A—Appendix A), which was taken in dry DMSO. For this reason, the signal is not blurred and occurs at a shift of 4.73 ppm. The analysis of the spectra confirmed the structures of the obtained compounds. No significant differences in chemical shifts were noticed. The spectra are analogous. Signal integrations confirm the ratio between the ester and betulinate portions in a given salt.

Figure 3 summarizes the ^13^C NMR spectra of betulinic acid and the obtained L-amino acid ethyl ester betulinates. Based on the analysis of the ^1^H and the ^13^C NMR spectra of all synthesized products, it was proved that the desired structures of compounds were obtained.

The identity of the BA-ILs was also confirmed based on analyses of ATR-FTIR spectra. All obtained L-amino acid ethyl ester salts have absorption bands in the wavelength range of about 1235 cm^−1^ and 1750 cm^−1^, corresponding to the stretching vibrations of the C–O and C=O groups, respectively. The appearance of these bands confirms the presence of the α-amino acid. The ATR-IR analysis revealed the presence of distinct absorption bands at approximately 1620 and 1390 cm^−1^, corresponding to symmetric ν(COO^−^)sym and asymmetric ν(COO^−^)as stretching vibrations, respectively. The confirmation of the carboxylate anion COO^−^ is evident from the observed difference exceeding 200 cm^−1^ between the frequencies of ν(COO^−^)sym and ν(COO^−^)as [65,66,67]. The band at about 2800–3000 cm^−1^ corresponds to overlapping bands of asymmetric vibrations stretching C-H and N–H. Figure 4 shows the ATR-FTIR spectra of the BA-ILs. Individual FTIR spectra are summarized in the supplement.

The physicochemical properties, such as melting point, thermal stability, and specific rotation, were determined for all obtained ILs. Table 1 summarizes the results.

Based on TG experiments, the stability of L-amino acid betulinates was evaluated. All TG and DTG curves are shown in the Appendix A. All obtained compounds decomposed in many stages. The least stable of the tested compounds was [AlaOEt][BA] (T_TGonset_ = 92.0 °C), and the most stable was [TrpOEt][BA] (T_TGonset_ = 231.2 °C), which was less stable than unmodified betulinic acid (T_TGonset_ = 305.7 °C). The observed differences in thermal stability among the tested compounds can be attributed to several factors related to the chemical structure and interactions within the compounds. At first, the choice of amino acid in the ILs can significantly influence the thermal stability of the resulting compounds. In this case, the least stable compound, [AlaOEt][BA], contains alanine, which has a relatively short side chain and may not stabilize the overall structure more than larger or more polar amino acids. On the other hand, the most stable compound, [TrpOEt][BA], contains tryptophan, which has a larger and more complex aromatic side chain that could contribute to greater stability through π–π stacking interactions. Intermolecular interactions between the cation and anion components also influence the thermal stability of ILs. Stronger interactions, such as hydrogen bonding or π–π stacking, can enhance the overall stability of the compound. The specific arrangement and strength of these interactions may vary depending on the nature of the amino acid and its compatibility with BA. Moreover, the presence of the ethyl ester group in the cation component of the ILs may also affect thermal stability. Ester groups can contribute to increased flexibility or mobility within the molecule, potentially impacting the overall stability of the compound. Comparing the thermal stability of the ILs to unmodified BA provides insights into how the addition of amino acids and the formation of ILs affect the compound’s overall stability. While some ILs may exhibit reduced stability compared to unmodified BA, this decrease may be acceptable considering the potential benefits of the improved solubility and bioavailability conferred by the IL formation.

The melting points of the derivatives obtained, where determined, were below 100 °C, which allows them to meet one of the conditions for classification into the group of ionic liquids.

Since optically active amino acids were utilized during synthesis, the optical rotation of the resulting derivatives was also investigated (Table 1). All obtained derivatives exhibited the ability to rotate the polarized light plane. All obtained betulinates rotated the plane of polarized light in the same direction as the starting BA.

The conjugation with L-lysine esters significantly affected the water solubility of poorly soluble betulinic acid [59]. In general, the solubility of salts derived from hydrophilic amino acids was also improved. In contrast, the solubility of derivatives obtained from aliphatic or branched amino acids was comparable to unmodified BA (Table 1).

### 3.2. Cytotoxic Effect of [AAOEt][BA] on MCF-7 Cells

To investigate how the conversion of BA to ILs affects its cytotoxicity toward MCF-7 cells, we used a standard MTT-dye reduction assay. The IC_50_ values (all in the micromolar range), calculated as the concentrations of the tested compounds causing 50% cellular toxicity, are shown in Table 2. The estimated IC_50_ (72 h) value for BA (11.5 ± 1.8 µM) is in agreement with those reported in the literature for MCF-7 cells [68,69]. One of the biggest exceptions to this trend was observed for [ValOEt][BA] (one of the branched amino acids with poor solubility shown in Table 1), which showed a statistically two-fold higher IC_50_ value compared to that of BA (25.7 ± 0.8 µM *, Table 2). The most cytotoxic BA formulation appeared to be [LysOEt][BA]_2_ (IC_50_ 4.8 ± 1.3 µM ***, Table 2), which is also one of the most soluble compounds according to Table 1. Interestingly, the monosubtituted L-LysOEt salt showed significantly lower toxicity (IC_50_ 12.9 ± 0.3 µM) than BA, regardless of its highest solubility (Table 1). Although we could not find a direct correlation between the water solubility of BA-ILs and their cytotoxicity, the contribution of the enhanced content of BA in the formulation for its cytotoxic effect is clearly visible. It is a well-known fact that the majority of eukaryotic cells exhibit a negatively charged cell membrane at physiological pH [70]. In this regard, the contribution of the cation (from the IL formulation) to the initial adsorption of [LysOEt][BA]_2_ to the negatively charged cancer cell membrane via electrostatic interactions and its further permeation should also be considered [71].Our results are in agreement with the work of Drag-Zalesinska et al., who showed that the covalently Lys-modified betulin and betulinic acid had one of the highest toxicities toward pancreatic cancer cells compared to other AA-betulin or AA-BA conjugates [60]. It is worth mentioning that derivatives with aromatic AA ([PheOEt][BA], [TyrOEt][BA], and [TrpOEt][BA]) showed cytotoxicity similar to BA, since derivatives containing AA with aliphatic hydrophobic side chains (especially the longer ones) have IC_50_ values lower than that of BA ([LeuOEt][BA], IleOEt][BA], and [MetOEt][BA]) (Table 2). Based on these results, we can assume that the type and length of the side chain of AA seems to have a major effect on cytotoxicity, as was referred to earlier [72,73,74]. We observed a positive correlation between toxicity and IL concentration and cation chain length, similar to that reported for 1, 3-dialkylimidazolium salts [75]. Since several studies have highlighted the relationship between IL toxicity and the degree of IL-cation lipophilicity, this suggests membrane damage as a potential cause of cytotoxicity [76,77,78]. We performed an MTT test in order to check the degree of cytotoxicity of [LysOEt][BA]_2_ (the most cytotoxic IL for cancer cells) to the non-cancerous cell line MCF-10A (Table 2). Our results did not show any cell specificity of action of the IL. The obtained IC_50_ for MCF-10A cells treated with ([LysOEt][BA]_2_) did not differ significantly from those found for MCF-7 (Table 2). The results are not surprising, taking into account that MCF-10 cells have an even greater negative charge on their membranes (−31.16 ± 1.12 mV) than MCF-7 cells (−20.32 ± 2.43 mV) [79]. Similar results were obtained for SH-479 derivative of BA as the cell viability of MCF-10A cells was in the range of 80% and above, only for low concentrations of the derivative (0.1–10 μM). If non-cancerous cells were treated with IC_50_ for MCF-7 cells (around 25 μM), the cell viability of MCF-10a cells would also drop down [80]. Therefore, such compounds should be used with caution in anti-tumor therapy.

### 3.3. Effect of BA and Two Selected [AAOEt][BA] on the Clonogenic Activity of MCF-7 Cells

Next, we compared the effect of two selected betulinates, [ValOEt][BA] (the least cytotoxic) and [LysOEt][BA]_2_ (the most cytotoxic), and the parent BA on the colony expansion capability of MCF-7 cells (Figure 5A). The survival fraction was determined by comparing the clonogenicity of the treated cells with that of the control (non-treated cells) (Figure 5B). The tested concentrations of the compounds were chosen in the range between 2.5 and 12.5 µM.

The dose-survival curve (Figure 5B) showed the ability of a single cell to grow into a large colony, indicating that it has retained its ability to proliferate. BA and [ValOEt][BA] did not inhibit MCF-7 colony formation and thus had little effect on the total number of surviving fractions. In contrast, treatment with the same increasing concentrations of [LysOEt][BA]_2_ reduced cell clonogenicity to below 20% surviving fraction at 12.5 µM (corresponding to approx. the IC_50_ of BA) (Figure 5B, Table 2). These data are in agreement with the results obtained for the cytotoxicity of the tested derivatives.

### 3.4. Cell Morphology Analysis

Using light microscopy, we monitored changes in cell numbers and morphology in MCF-7 cells treated for 72 h with the least and the most toxic BA-ILs, i.e., [ValOEt][BA] and [LysOEt][BA]_2_, and compared the results with those for BA-treated and untreated MCF-7 cells (Figure 6). The results showed that treatment of MCF-7 cells with BA at a concentration close to its IC_50_ value mainly caused a reduction in cell numbers compared to the control cells. The same effect was observed for [ValOEt][BA]. These data are consistent with the results obtained for both compounds from the cytotoxicity (Table 2) and clonogenic capacity assays (Figure 5).

On the other hand, incubation of MCF-7 cells with [LysOEt][BA]_2_ led not only to decreased cell numbers as a consequence of reduced proliferative activity and clonogenic capacity (Table 2 and Figure 5), but also caused morphological changes such as cell rounding and shrinkage (Figure 6).

### 3.5. Thermodynamic Behavior of MCF-7 Cells Treated with BA or [LysOEt][BA]_2_

We investigated the effect of BA and [LysOEt][BA]_2_, the compound that showed the highest cytotoxicity and suppressed clonogenicity, on the thermal denaturation of MCF-7 cells after 72 h of treatment. The DSC profile of the non-treated MCF-7 cells showed six endothermic thermal transitions in the temperature range of 30–120 °C (designated as T_1_ to T_6_) and two shoulders (Sh_1_ and Sh_2_) (Figure 7). The transition temperatures (Tm) and excess heat capacity for the major peaks are listed in Table 3. The peaks in the low temperature (L_T_) region (around 50–73 °C) are attributed to non-chromosomal components such as cytosolic proteins and RNA, while the peaks in the high temperature (H_T_) region (above 75 °C) are mainly assigned to the denaturation of the nuclear components [81,82,83].

The DSC curves of MCF-7 cells treated for 72 h with BA and [LysOEt][BA]_2_ at IC_50_ concentrations did not differ significantly with regard to the temperature’s midpoint from the corresponding transitions of the control (non-treated MCF-7 cells). However, it should be noted that the main transition in the L_T_ region (T_1_) of MCF-7 treated with BA was downshifted by more than 1 °C with concomitant diminished amplitude compared to the control (Figure 7, Table 3). The same tendency was also observed for T_3_ thermal transition (Figure 7, Table 3). There was no significant difference in the thermal profiles of all tested conditions in the rest of the L_T_ region. All transitions in BA- and [LysOEt][BA]_2_-treated cells exhibited lower c_Pex_ in the L_T_ region compared to the control cells (Figure 7, Table 3). As shown in Table 3, the effect of BA is more pronounced than that of IL-treated cells. In summary, the observed alterations in the thermal transitions and heat capacity values in the L_T_ region of BA- and [LysOEt][BA]_2_-treated cells suggest an effect on cytosol components at the tested concentrations. Similarly, another study has shown that some 1, 3-dialkylimidazolium- or tetraalkylphosphonium-based ionic liquids can affect the cytoplasmic proteins and enzymes, RNA, ribosomes, and/or mitochondria of cancer cells [47,84].

Regarding the H_T_ region, we noticed that the treatment of MCF-7 cells with BA and [LysOEt][BA]_2_ had a subtle effect on the thermograms (Figure 7, Table 3). Typically, three thermal transitions can be clearly distinguished in this region, which is associated with nucleosome collapse, melting of the relaxed DNA released from the nucleosome, and melting of the released intact supercoiled DNA [76]. For instance, significant shifts in the thermal transitions in the H_T_ regions of thermograms of cancer cells treated with strong intercalators and DNA strand-disrupting anticancer drugs (ethidium bromide, actinomycin D, or doxorubicin), alkylating agents (chlorambucil, carboplatin, or cisplatin), or UV light have been reported in the literature [83,85]. Our data showed slight but noticeable changes in the thermal transitions of the H_T_ region only for BA-treated MCF-7 cells (Figure 7, Table 3). Moreover, we observed a decreased T_5_ and T_6_ excess heat capacity for both treatments, especially for [LysOEt][BA]_2_. This is consistent with our results demonstrating a greater cytotoxic effect and suppressed clonogenicity formation for the IL-modified BA, and the reported BA-induced DNA damage in cancer cells [86,87].

In the thermograms, the lower temperature shoulder (Sh_1_) is probably due to the overlapping thermal events of some of the lipids comprising the cellular membrane [88,89]. Here, we observed the most notable changes. This transition was downshifted in the BA-treated MCF-7 and [LysOEt][BA]_2_ cells by 6 °C with respect to the control cells. Again, a stronger effect was observed for the IL-modified BA. It is noteworthy that a downshift in the Tm characteristic of phospholipids was observed in a study of interactions between a model of the cell membrane and ILs containing hydrophobic cations [90]. The obtained data agree with the reported change in the structural stability of biomembranes in the presence of low ILs concentrations [91]. The proposed mechanism of destabilization of the lipid membrane involves an electrostatic interaction between the IL-cation and a negatively charged oxygen atom of phospholipids, as well as hydrophobic interactions between an aromatic moiety of either cation or anion and the heads and tails of phospholipids [47]. In addition, Kumari et al. found that with respect to the hydrophobicity of the cation, ILs tested at sub-toxic concentrations may affect the cell migration of MDA-MB-231 by altering the membrane elasticity and penetration resistance of the cellular lipid membrane [91]. A decrease in the membrane rigidity of MDA-MB-231 cells treated with imidazolium-based ILs was reported by other authors as well [92]. Therefore, alteration in the mechanical properties such as the elasticity, viscosity, and/or permeability of the cell membrane is a possible explanation for the observed effects on the thermal profiles in the H_T_ region.

## 4. Conclusions

We proposed an optimized procedure for the preparation of L-amino acid ethyl ester betulinates. As a result, 15 new ILs were synthesized, and structurally and physicochemically characterized. The incorporation of a cation of charged or polar AAs improved the water solubility of BA. Within the [AAOEt][BA] series, the most pronounced cytotoxic effect toward MCF-7 cells was observed for [LysOEt][BA]_2_. We could suggest that the charged AA provokes cytotoxicity by a mechanism involving membrane damage leading to alternation in lipid organization and/or modulation of the visco-elastic environment of the cytoplasm.

This pilot study demonstrates that the conversion of BA into IL with esters of AAs can be a reliable strategy to achieve improved cytotoxicity or target a specific cellular compartment and increase the efficacy of BA for practical medical application.

## Figures and Tables

**Figure 1 pharmaceutics-16-00496-f001:**
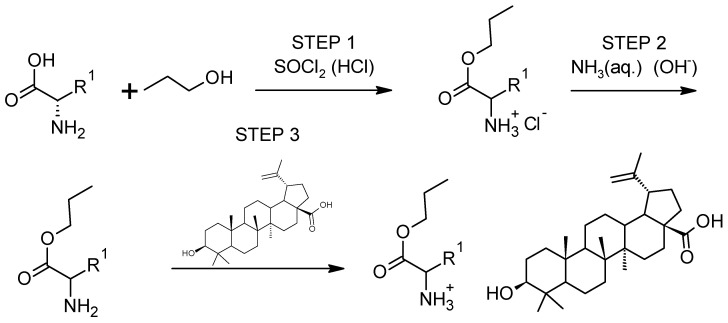
General synthesis path of [AAOEt][BA].

**Figure 2 pharmaceutics-16-00496-f002:**
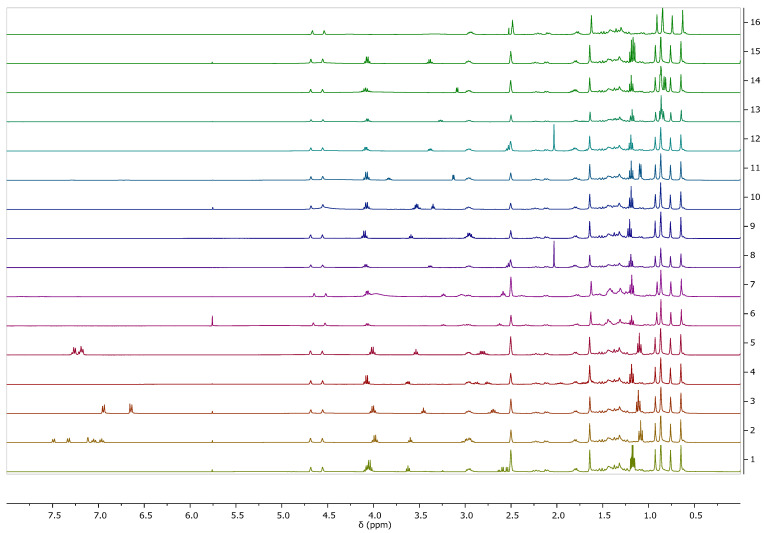
^1^H NMR spectra for betulinic acid and L-amino acid ethyl ester salts in the range 0–5 ppm, from the top: BA, [AlaOEt][BA], [ValOEt][BA], [LeuOEt][BA], [IleOEt][BA], [ThrOEt][BA], [SerOEt][BA], [CysOEt][BA], [MetOEt][BA], [LysOEt][BA], [LysOEt][BA]_2_, [PheOEt][BA], [ProOEt][BA], [TyrOEt][BA], [TrpOEt][BA], and [Asp(OEt)_2_][BA]. Individual proton NMR spectra are summarized in the supplement.

**Figure 3 pharmaceutics-16-00496-f003:**
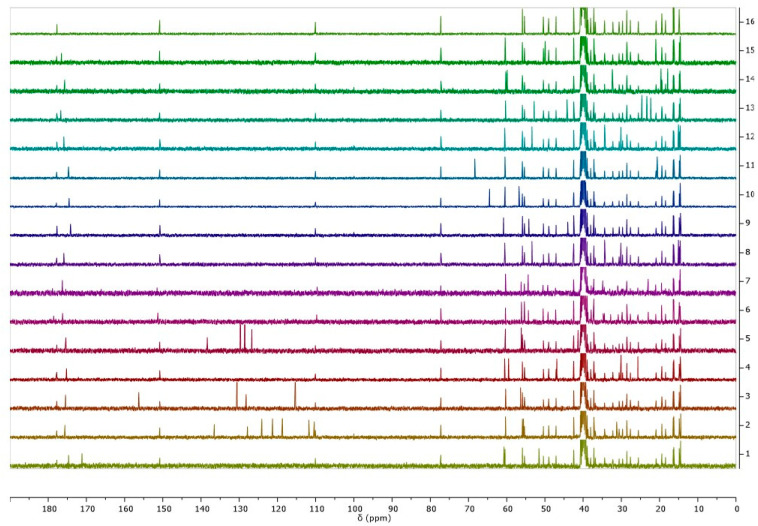
^13^C NMR spectra for betulinic acid and L-amino acid ethyl ester salts in range 0–190 ppm, from the top: BA, [AlaOEt][BA], [ValOEt][BA], [LeuOEt][BA], [IleOEt][BA], [ThrOEt][BA], [SerOEt][BA], [CysOEt][BA], [MetOEt][BA], [LysOEt][BA], [LysOEt][BA]_2_, [PheOEt][BA], [ProOEt][BA], [TyrOEt][BA], [TrpOEt][BA], and [Asp(OEt)_2_][BA]. Individual carbon NMR spectra are summarized in the supplement.

**Figure 4 pharmaceutics-16-00496-f004:**
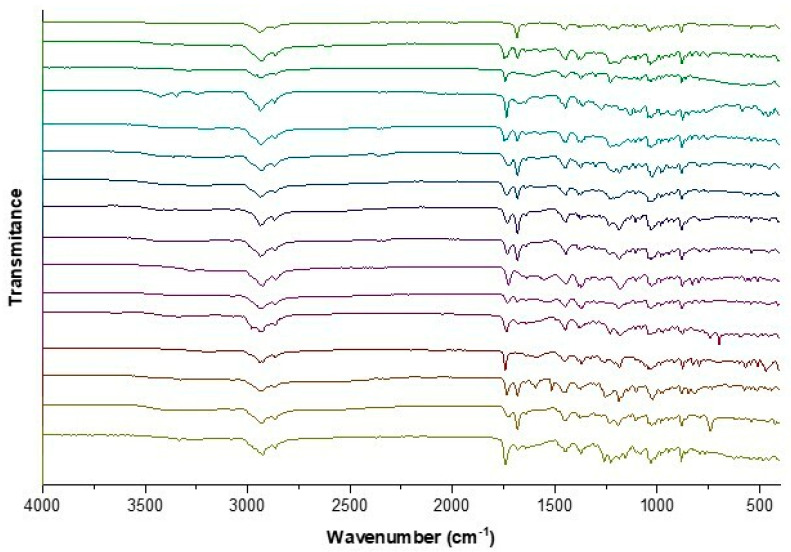
ATR-FTIR for betulinic acid and L-amino acid ethyl ester salts, from the top: BA, [AlaOEt][BA], [ValOEt][BA], [LeuOEt][BA], [IleOEt][BA], [ThrOEt][BA], [SerOEt][BA], [CysOEt][BA], [MetOEt][BA], [LysOEt][BA], [LysOEt][BA]_2_, [PheOEt][BA], [ProOEt][BA], [TyrOEt][BA], [TrpOEt][BA], and [Asp(OEt)_2_][BA].

**Figure 5 pharmaceutics-16-00496-f005:**
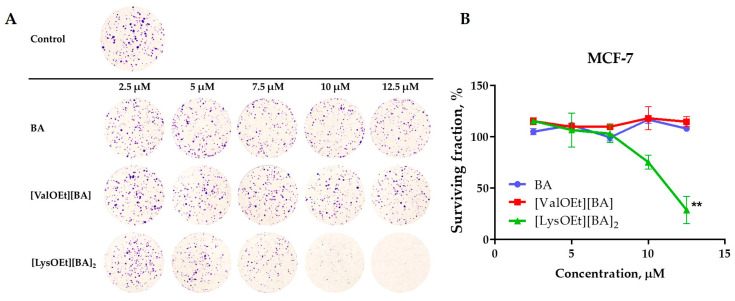
(**A**) Clonogenic assay of MCF-7 cells treated with the indicated BA or [AAOEt][BA] concentrations; (**B**) Dose—survival curve visualizes the mean surviving fractions ± SEM of MCF-7 cells treated with the indicated BA or [AAOEt][BA] concentrations. Student *t*-test—** *p* < 0.01 of [LysOEt][BA]_2_ vs. BA and [ValOEt][BA]; (n = 3).

**Figure 6 pharmaceutics-16-00496-f006:**
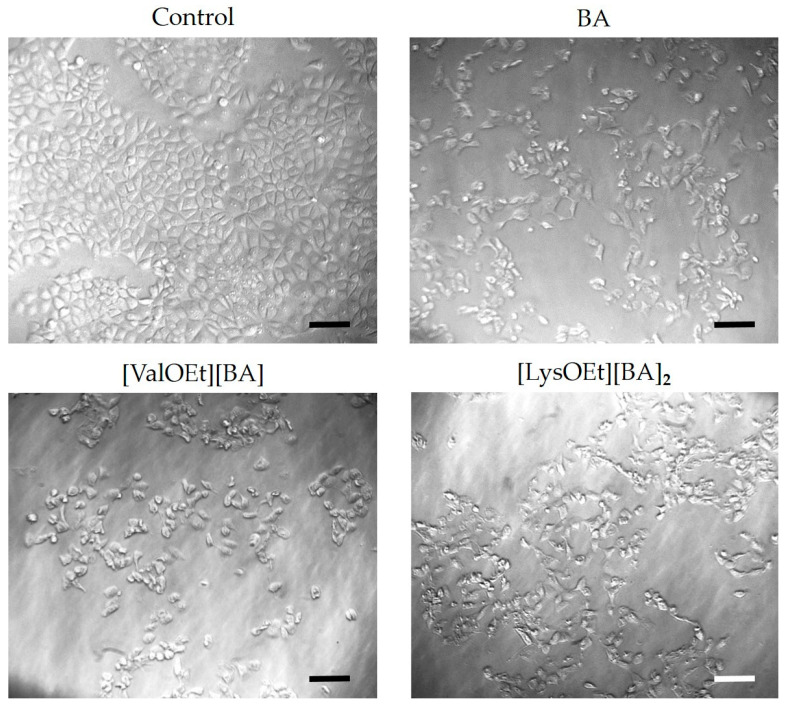
Light-microscopy images of non-treated MCF-7 cells (control) and MCF-7 cells treated with BA, [ValOEt][BA], and [LysOEt][BA]_2_ for 72 h. Scale bar—100 µm.

**Figure 7 pharmaceutics-16-00496-f007:**
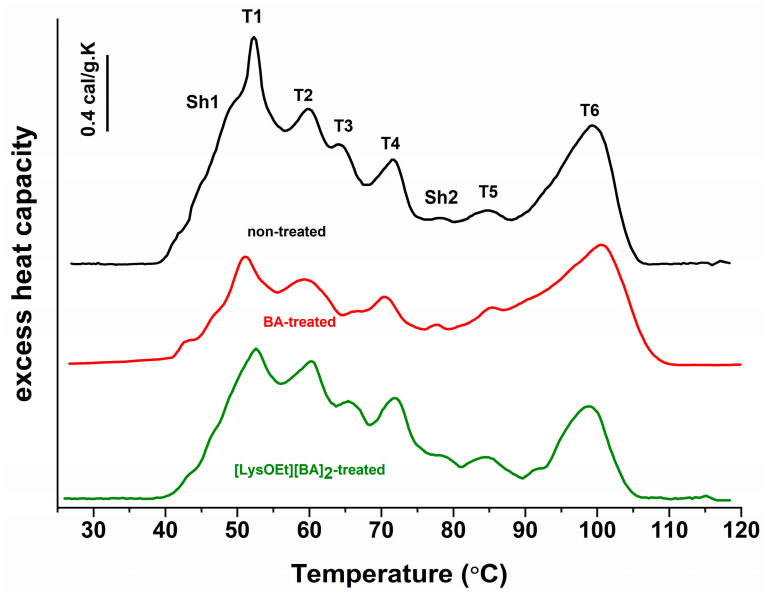
DSC curves of non-treated MCF-7 cells (black) (control) and MCF-7 cells treated for 72 h with 11.5 µM BA (red) or 4.8 µM BA (red) or [LysOEt][BA]_2_ (green). The successive transitions in the control curve are denoted by T_1_–T_6_ and the shoulders by Sh_1_ and Sh_2_. Representative curves from each condition were obtained from two independent experiments.

**Table 1 pharmaceutics-16-00496-t001:** Selected properties of BA and [AAOEt][BA].

Compound	T_m_ (°C)	T_TGonset_ (°C)	T_DTGmax_ (°C)	${\left[\alpha \right]}_{\lambda }^{20}$	${\left[M\right]}_{\lambda }^{20}$	Water Solubility
BA	-	305.7	358.7	+8.841	+41.173	weak
[AlaOEt][BA]	-	92.0	370.7	+8.478	+49.412	weak
[ValOEt][BA]	-	116.8	354.0	+15.101	+92.252	weak
[LeuOEt][BA]	-	114.9	359.4	+9.654	+60.331	weak
[IleOEt][BA]	88.8	127.5	369.0	+6.229	+38.927	weak
[ThrOEt][BA]	51.5	109.0	353.5	+7.512	+46.038	middle
[SerOEt][BA]	-	118.2	364.1	+7.020	+42.040	middle
[CysOEt][BA]	-	150.1	359.2	+21.818	+134.165	middle
[MetOEt][BA]	86.5	125.7	358.6	+7.420	+47.708	weak
[LysOEt][BA]	84.9	144.0	327.3	+11.091	+70.976	very good
[LysOEt][BA]_2_	-	157.7	368.8	+10.413	+236.367	good
[PheOEt][BA]	82.8	133.9	360.1	+15.332	+101.032	middle
[ProOEt][BA]	91.2	121.6	372.1	+2.235	+13.607	weak
[TyrOEt][BA]	70.7/103.1	202.3	354.4	+15.810	+106.712	weak
[TrpOEt][BA]	-	231.3	359.8	+15.436	+107.743	middle
[Asp(OEt)_2_][BA]	76.5	119.8	357.4	+4.057	+26.569	middle

T_m_—melting maximum peak temperature, T_TGonset_—the onset of the thermal degradation, T_DTGmax_—maximum decomposition temperature, ${\left[\alpha \right]}_{\lambda }^{20}$—specific rotation, ${\left[M\right]}_{\lambda }^{20}$—molar rotation, and categories of solubility—“5×”: weak; “10×”: middle; “20×”:good’; “50×”: very good; and “100×”: excellent.

**Table 2 pharmaceutics-16-00496-t002:** IC_50_ values (72 h) of BA and [AAOEt][BA] on MCF-7 and MCF-10A cells.

Compound	IC_50_, µM
BA	11.5 ± 1.8
[AlaOEt][BA]	12.7 ± 4.5
[ValOEt][BA]	25.7 ± 0.8 *
[LeuOEt][BA]	7.5 ± 1.2
[IleOEt][BA]	8.5 ± 1.3
[ThrOEt][BA]	8.2 ± 2.4
[SerOEt][BA]	12.2 ± 2.6
[CysOEt][BA]	9.6 ± 3.8
[MetOEt][BA]	7.5 ± 0.3
[LysOEt][BA]	12.9 ± 0.3
[LysOEt][BA]_2_	4.8 ± 1.3 *** (4.0 ± 0.2 *** for MCF-10A)
[PheOEt][BA]	12.3 ± 1.7
[ProOEt][BA]	7.6 ± 1.8
[TyrOEt][BA]	14.2 ± 2.4
[TrpOEt][BA]	13.1 ± 0.4
[Asp(OEt)_2_][BA]	8.9 ± 2.1

Each determined IC_50_ value of [AAOEt][BA] was compared to the IC_50_ of BA using one-way ANOVA. * *p* < 0.05 and *** *p* < 0.001; (n = 3).

**Table 3 pharmaceutics-16-00496-t003:** Thermodynamic parameters (transition temperature, Tm (°C), excess heat capacity, and c_P_^ex^ (cal/g·K)) of the successive thermal transitions for non-treated (control) and treated with BA or [LysOEt][BA]_2_ MCF-7 cells.

Thermodynamic Parameters	MCF-7	MCF-7-BA	MCF-7-[LysOEt][BA]_2_
Sh_m1_ (°C)	48.9	46.80	43.4
c_P_^Sh1^ (cal/g·K)	0.83	0.13/	0.14
T_m1_ (°C)	52.4	51.2	52.7
c_P_^T1^ (cal/g·K)	1.21	0.58	0.80
T_m2_ (°C)	59.8	59.2	60.2
c_P_^T2^ (cal/g·K)	0.83	0.45	0.73
T_m3_ (°C)	64.2	66.3 (Sh)	65.5
c_P_^T3^ (cal/g·K)	0.64	0.29	0.52
T_m4_ (°C)	71.5	70.5	71.8
c_P_^T4^ (cal/g·K)	0.55	0.36	0.54
Sh_m2_ (°C)	78.3	77.6	78.4
c_P_^Sh2^ (cal/g·K)	0.24	0.22	0.23
T_m5_ (°C)	84.8	85.5	84.4
c_P_^T5^ (cal/g·K)	0.28	0.31	0.22
T_m6_ (°C)	99.2	100.7	98.9
c_P_^T6^ (cal/g·K)	0.73	0.64	0.49

## Data Availability

The data presented in this study are available in this article and Appendix A.

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
