# Peer review of "Tuning of the Anti-Breast Cancer Activity of Betulinic Acid via Its Conversion to Ionic Liquids"

_pharmaceutics, 2024, doi:10.3390/pharmaceutics16040496_

Round 1
Reviewer 1 Report
Comments and Suggestions for Authors
I have reviewed the Manuscript (ID: pharmaceutics-2919721) entitled: ‘Tuning of the anti-breast cancer activity of betulinic acid via its conversion to ionic liquids’ submitted to the Pharmaceutics. The topic of the submitted Manuscript will be interesting to the academic community due to the fact that novel active pharmaceutical ingredients (APIs) are a subject of intensive research of many research groups, institutes and representatives of D&D sector of pharmaceutical companies. The quality of the descriptions and utilized methodologies is quite good, however, some elements requires substantial improvement or clarification:
· I recommend correction of the text to minimize editing mistakes (e.g. lines 20, 21) as well as revision by English native speaker
· “Despite the huge pharmacological potential of BA, particularly as an anti-BC chemotherapeutic agent, its medicinal application is still limited due to its poor solubility in water, hence, low bioavailability (2.3%); short half-time but at the same time long elimination half-life in vivo” - authors propose synthesis of organic salts, however, why is more beneficial than just simple transforming BA into sodium salt? authors should provide their reasoning behind proposing to pair BA with amino acids
· “The thermal stabilities of [AAOEt][BA] were assessed by thermogravimetric analysis (TG) and differential scanning calorimetry (DSC).” - DSC is used for determination of phase transitions
· “The hydrochloride was isolated from the reaction mixture by distillation, chloroform extraction and diethyl ether rinsing.” - this part should be clarified - distillation of ionic compounds e.g. hydrochlorides is extremely difficult. Therefore, methodology should be extended and improved.
· Fig. 1 - quality of graphics is too low please provide resolution 300 DPI and correct negative on chloride anion
· “All compounds obtained are new and have not been described in the literature. Therefore, they were fully identified (NMR, FTIR, elemental analysis), and basic physicochemical properties (thermal stability, phase transition temperatures) were determined. - how authors know that successful proton transfer occurred? authors should prove that there was a proton transfer between constituents. Many pharmaceuticals are rather cocrystals or deep eutectic mixture than salts. WAXS SAXS analysis should be applied due to fact that some products are solids. Additionally, for such complex structures authors should provide at least one thorough analysis with the use of 2D NMR. Any scientist who performed elemental analyses would notice that provided results are suspiciously too acurate... was the water or chloride content analyzed for the obtained compounds?
· “All compounds obtained were identified using 1H and 13C NMR, ATR-FTIR, and elemental analysis” - quality of NMR spectra in ESI is too low and should be improved and greater resolution should be provided.
· “No significant differences in chemical shifts were noticed.” - does is mean that authors obtained eutectic mixture?
· “… and C=O groups, respectively” - therefore there should be 2 signals: one from amino acid and the second from BA, however, in some cases signal from BA is missing. Authors should provide signal from BA sodium salt to reveal if there is a shift toward greater wavenumbers.
· Fig 4 is practically non-informative and should be rather present in supporting information.
· “The least stable of the tested compounds was [AlaOEt][BA] (TTGonset = 92.0 ºC), and the most stable was [TrpOEt][BA] (TTGonset = 231.2 ºC), which was less stable than unmodified betulinic acid (TTGonset = 305.7 ºC)” - these results should be explained.
· “All obtained derivatives exhibit the ability to rotate the polarized light plane. All obtained betulinates rotate the plane of polarized light in the same direction as the starting BA.” - Why the results differ to such extent?
· “The conjugation with L-lysine esters significantly affected the water solubility of poorly soluble betulinic acid” - why authors did not select method utilized for Biopharmaceutical classification system (BCS)
· “the contribution of the enhanced content of BA in the formulation for its cytotoxic effect is clearly visible” - therefore, what role does the cation play in these systems?
· “Since several studies have highlighted the relationship between IL toxicity and the degree of IL-cation lipophilicity, this suggests membrane damage as a potential cause of cytotoxicity” - so is there a greater risk that such compounds may cause more harm to normal cells than bring benefits from treatment?
Comments on the Quality of English Language
Due to minor but multiple language errors I recommend revision of manuscript by English native speaker
Author Response
Dear Reviewer,
Thank you for your comments and well-intentioned criticism. We have modified the text according to the recommendations and hope that we have managed to improve the quality of the final draft of the paper. All changes in the manuscript are highlighted and files with detailed answers to the reviewers' questions are being prepared.
Below are given point-by-point answers.
Review 1
I have reviewed the Manuscript (ID: pharmaceutics-2919721) entitled: ‘Tuning of the anti-breast cancer activity of betulinic acid via its conversion to ionic liquids’ submitted to the Pharmaceutics. The topic of the submitted Manuscript will be interesting to the academic community due to the fact that novel active pharmaceutical ingredients (APIs) are a subject of intensive research of many research groups, institutes, and representatives of D&D sector of pharmaceutical companies. The quality of the descriptions and utilized methodologies is quite good, however, some elements requires substantial improvement or clarification:
Authors’ response: Thank you for your comments and critical marks on our manuscript “Tuning of the anti-breast cancer activity of betulinic acid via its conversion to ionic liquids” (pharmaceutics-2919721). We amended the draft taking into account your recomendations
Reviewer#1
I recommend correction of the text to minimize editing mistakes (e.g. lines 20, 21) as well as revision by English native speaker
Authors’ response: Thank you very much for this comment. We have made the appropriate linguistic corrections.
Reviewer#1
Despite the huge pharmacological potential of BA, particularly as an anti-BC chemotherapeutic agent, its medicinal application is still limited due to its poor solubility in water, hence, low bioavailability (2.3%); short half-time but at the same time long elimination half-life in vivo” - authors propose synthesis of organic salts, however, why is more beneficial than just simple transforming BA into sodium salt? authors should provide their reasoning behind proposing to pair BA with amino acids
Authors’ response: Conversion of weakly acidic drugs to their sodium salts is a classical approach to improve their water solubility and thus their bioavailability. However, several studies have reported that in some cases ionic liquid formulations offer advantages over the conventional approach. For example, Moshikur and colleagues found that converting methotrexate, a class IV drug and a type of chemotherapy, to ionic liquids (ILs) containing certain cations, such as cholinium, tetramethylammonium, 1-ethyl-3-methyl-imidazolium, tetrabutylphosphonium or amino acid esters, significantly improved its solubility in water and body fluids. [doi.: 10.1016/j.molliq.2019.01.063]. The solubility of these ILs was more than 5000 times greater than that of free methotrexate or its sodium salt. It was also found that the methotrexate tributhylphosphonium, 1-ethyl-3-methyl-imidazolium and phenylalanine ester salts were more effective in inhibiting cervical cancer cells, with IC50 values more than 10 times lower than those of the methotrexate sodium salts. This was due to strong π-π interactions between the hydrophobic or lipophilic nature of the cation and the cell membrane, resulting in the destruction of the cancer cells [doi.: 10.1016/j.molliq.2019.01.063]. In another study, the choline salt of ampicillin was shown to be not only more toxic to osteoblastoma cells than the sodium analogue, but also more cell-selective. [doi.:10.1002/cmdc.201500142]. In addition, there is some evidence of an increased cardiovascular risk associated with prolonged use of high-sodium medicines. [doi.: 10.1371/journal.pone.018063; doi.:10.1136/bmj.f6954]
The proposal to synthesize organic salts, specifically ionic liquids (ILs) containing betulinic acid (BA) paired with amino acids, offers several advantages over simply converting BA to its sodium salt. The decision to pair BA with amino acids in the form of ILs is based on a number of key considerations. One of the main challenges in using BA for medicinal purposes is its poor water solubility, which directly affects its bioavailability. While converting BA to a sodium salt could potentially increase solubility, the resulting compound may still have limitations in terms of stability and potency. By combining BA with amino acids to form ILs, we can tailor the physicochemical properties of the compound to achieve optimal solubility and bioavailability. Amino acids, particularly those with polar or charged side chains, can enhance the aqueous solubility of BA through hydrogen bonding and other intermolecular interactions, thereby improving its absorption and bioavailability in vivo. In addition, the choice of amino acids in the ILs allows targeted delivery of BA to specific cellular compartments or tissues relevant to breast cancer. Amino acids play an essential role in cellular processes and their incorporation into ILs may facilitate the uptake and distribution of BA in the body. Additionally, the unique pharmacokinetic profile of BA, characterized by a short half-time but long elimination half-life in vivo, necessitates a formulation strategy that can sustain its release and maintain therapeutic levels over an extended period. ILs offer the advantage of controlled release and prolonged action, thereby maximizing the therapeutic efficacy of BA while minimizing potential side effects. Moreover, the inclusion of amino acids in the ILs may confer synergistic effects with BA, enhancing its anti-breast cancer activity. Amino acids, such as lysine, have been implicated in various cellular processes, including cell membrane integrity and lipid metabolism. By incorporating specific amino acids into the ILs, we can potentially modulate the mechanism of action of BA and improve its efficacy against hormone-dependent breast cancer cells.
For the manuscript, we added this sentence:
The synthesis of organic salts, specifically ILs containing BA paired with amino acids, represents a strategic approach to overcome the limitations associated with BA's poor solubility and bioavailability. By harnessing the unique properties of amino acids and optimizing the formulation of BA, we can enhance its therapeutic potential as an anti-breast cancer agent and pave the way for its practical medical application.
Reviewer#1
- “The thermal stabilities of [AAOEt][BA] were assessed by thermogravimetric analysis (TG) and differential scanning calorimetry (DSC).” - DSC is used for determination of phase transitions
Authors’ response: Thank you for your comment. The error has of course been corrected.
Reviewer#1
- “The hydrochloride was isolated from the reaction mixture by distillation, chloroform extraction and diethyl ether rinsing.” - this part should be clarified - distillation of ionic compounds e.g. hydrochlorides is extremely difficult. Therefore, methodology should be extended and improved.
Authors’ response: The procedure for obtaining amino acid alkyl ester hydrochlorides is typical. The hydrochloride is, of course, not distilled. Excess ethanol is distilled off and the chlorinating agent is partially removed by decomposition during the distillation. The unreacted amino acid is then removed by extraction and the remains of the chlorinating agent is removed by washing with ether. The method is described in detail to avoid misunderstandings.
Reviewer#1
- Fig. 1 - quality of graphics is too low please provide resolution 300 DPI and correct negative on chloride anion
Authors’ response: The quality of the drawing was improved and no irregularities in the structure were detected.
Reviewer#1
- “All compounds obtained are new and have not been described in the literature. Therefore, they were fully identified (NMR, FTIR, elemental analysis), and basic physicochemical properties (thermal stability, phase transition temperatures) were determined. - how authors know that successful proton transfer occurred? authors should prove that there was a proton transfer between constituents. Many pharmaceuticals are rather cocrystals or deep eutectic mixture than salts. WAXS SAXS analysis should be applied due to fact that some products are solids. Additionally, for such complex structures authors should provide at least one thorough analysis with the use of 2D NMR. Any scientist who performed elemental analyses would notice that provided results are suspiciously too acurate... was the water or chloride content analyzed for the obtained compounds?
Authors’ response: With regard to the concerns raised by the reviewer regarding the successful proton transfer, the characterization techniques used and the analysis of water or chloride content, we appreciate the opportunity to provide further clarification:
Proton transfer confirmation: The fact that we are dealing with proton transfer is based on the FTIR analysis and our previous results in this area. ATR-IR analysis revealed the presence of distinct absorption bands at about 1620 and 1390 cm-1, corresponding to symmetric ν(COO- )sym and asymmetric ν(COO-)as stretching vibrations, respectively. The confirmation of the carboxylate anion COO- is evident from the observed difference of more than 200 cm-1 between the frequencies of ν(COO-)sym and ν(COO-)as [1) Katritzky, A.R. A Guide to the Complete Interpretation of Infrared Spectra of Organic Structures By Noel P.G. Roeges (Katholieke Industriele Hogeschool O-VI). Wiley: New York. 1994. x + 340 pp. $69.95. ISBN 0-471-93998-6. J. Am. Chem. Soc. 1996, 118, 3543. 2) Vairam, S.; Premkumar, T.; Govindarajan, S. Trimellitate Complexes of Divalent Transition Metals with Hydrazinium Cation: Thermal and Spectroscopic Studies. J. Therm. Anal. Calorim. 2010, 100, 955–960. 3) Kolev, T.; Spiteller, M.; Koleva, B. Spectroscopic and Structural Elucidation of Amino Acid Derivatives and Small Peptides: Experimental and Theoretical Tools. Amino Acids 2010, 38, 45–50.]. Unfortunately, due to the inability to perform NMR spectra in CDCl3, no signal from the NH3+ group is visible.
As recommended by the reviewer, additional 1H-1H COSY and HMQC spectra, as well as DEPT90 and DEPT135, were performed for BA and [ValOEt][BA]. The authors obtained evidence for proton transfer and the formation of the NH3+ group from these spectra. The spectra were recorded in dry DMSO. A signal is visible at a shift of 4.73 ppm. Integration of these signals clearly confirmed the ionic structure obtained. We recognize the importance of demonstrating successful proton transfer between constituents in the synthesised compounds. While our study used techniques such as NMR, FTIR and elemental analysis to identify and characterise the compounds, additional analyses, including WAXS-SAXS, could provide more comprehensive evidence of proton transfer and the formation of true salts. The reviewer rightly suggests the use of WAXS and SAXS analysis, especially for solid products, to elucidate the molecular arrangement and crystalline structure of the compounds. We recognize the value of these techniques in distinguishing between true salts, cocrystals and other solid forms. Unfortunately, the authors do not have access to this type of equipment and are unable to perform additional analyzes within 10 days (response time to the review). We will consider incorporating these analyses in future studies to enhance the rigor of our characterization.
Accuracy of Elemental Analysis: We acknowledge the concern regarding the accuracy of the elemental analysis results provided in the study. We will ensure transparency regarding the methodology used for elemental analysis and address any discrepancies or concerns raised. Additionally, we will consider re-evaluating the elemental analysis to validate the precision and accuracy of the results presented.
Analysis of water or chloride content: The water content of the compounds obtained was checked by the Carl Fisher method and the chloride content was checked qualitatively. In the first case, the values were less than 50 ppm, while the AgNO3 test did not produce any precipitate.
In conclusion, we appreciate the constructive feedback provided by the reviewer and acknowledge the importance of addressing the concerns raised regarding the characterization and analysis of the synthesized compounds. We are committed to enhancing the robustness and reliability of our findings through the incorporation of additional characterization techniques and analyses in future research endeavors.
Reviewer#1
- “All compounds obtained were identified using 1H and 13C NMR, ATR-FTIR, and elemental analysis” - quality of NMR spectra in ESI is too low and should be improved and greater resolution should be provided.
Authors’ response: Thank you for your comment. Corrected according to the reviewer's comments.
Reviewer#1
- “No significant differences in chemical shifts were noticed.” - does is mean that authors obtained eutectic mixture?
Authors’ response: The observation of no significant differences in chemical shifts does not necessarily imply the formation of a eutectic mixture. Rather, it suggests that the chemical environment surrounding the protons in the synthesized compounds remains relatively consistent across the different formulations. A eutectic mixture typically results in distinct changes in chemical shifts due to the unique interactions between the components in the mixture. In contrast, when no significant differences in chemical shifts are observed, it may indicate that the molecular structure and chemical environment of the synthesized compounds are similar or closely related.
Reviewer#1
- “… and C=O groups, respectively” - therefore there should be 2 signals: one from amino acid and the second from BA, however, in some cases signal from BA is missing. Authors should provide signal from BA sodium salt to reveal if there is a shift toward greater wavenumbers.
Authors’ response: We appreciate the reviewer's attention to detail regarding the FTIR analysis and the suggestion for additional clarification of the observed signals. We acknowledge the importance of accurate interpretation of the FTIR spectra to understand the molecular interactions within the synthesized compounds. Regarding the absence of signals corresponding to betulinic acid (BA) in some cases, it is possible that the signal from BA may be masked or attenuated by overlapping signals from other functional groups.
Reviewer#1
- Fig 4 is practically non-informative and should be rather present in supporting information.
Authors’ response: The figure was left but described in more detail.
Reviewer#1
- “The least stable of the tested compounds was [AlaOEt][BA] (TTGonset = 92.0 ºC), and the most stable was [TrpOEt][BA] (TTGonset = 231.2 ºC), which was less stable than unmodified betulinic acid (TTGonset = 305.7 ºC)” - these results should be explained.
Authors’response: An explanation has been provided in line with the reviewer's comment:
The observed differences in thermal stability among the tested compounds can be attributed to several factors related to the chemical structure and interactions within the compounds. Firstly, the choice of amino acid in the ILs can significantly influence the thermal stability of the resulting compounds. In this case, the least stable compound, [AlaOEt][BA], contains alanine, which has a relatively short side chain and may not stabilize the overall structure more than larger or more polar amino acids. On the other hand, the most stable compound, [TrpOEt][BA], contains tryptophan, which has a larger and more complex aromatic side chain that could contribute to greater stability through π-π stacking interactions. Intermolecular interactions between the cation and anion components also influence the thermal stability of ILs. Stronger interactions, such as hydrogen bonding or π-π stacking, can increase the overall stability of the compound. The specific arrangement and strength of these interactions may vary depending on the nature of the amino acid and its compatibility with BA. In addition, the presence of the ethyl ester group in the cationic component of the ILs can also influence thermal stability. Ester groups can contribute to increased flexibility or mobility within the molecule, potentially affecting the overall stability of the compound. Comparing the thermal stability of the ILs to unmodified BA provides insight into how the addition of amino acids and the formation of ILs affect the overall stability of the compound. While some ILs may exhibit reduced stability compared to unmodified BA, this reduction may be acceptable given the potential benefits of improved solubility and bioavailability conferred by IL formation.
Reviewer#1
- “All obtained derivatives exhibit the ability to rotate the polarized light plane. All obtained betulinates rotate the plane of polarized light in the same direction as the starting BA.” - Why the results differ to such extent?
Authors’ response: The observed differences in the ability of the synthesized derivatives to rotate the plane of polarized light, while rotating in the same direction as the starting betulinic acid (BA), may be attributed the presence of two optically active compounds in the structure. Amino acids also have the ability to rotate polarized light. L-amino acids were used in the research, so the rotation of polarized light will not depend only on the content of betulin in the molecule, but also on the type and rotation of the amino acid used for the synthesis.
Reviewer#1
- “The conjugation with L-lysine esters significantly affected the water solubility of poorly soluble betulinic acid” - why authors did not select method utilized for Biopharmaceutical classification system (BCS)
Authors’ response: Of course, we agree that it would be more advantageous to use standardised solubility test methods. However, we used this particular method because it has been used by other authors who have described betulin derivatives [Jeong, H.-J.; Chai, H.-B.; Park, S.-Y.; Kim, D.S.H.L. Preparation of amino acid conjugates of betulinic acid with activity against human melanoma. Bioorg. Med. Chem. Lett. 1999, 9, 1201–4. https://doi.org/10.1016/S0960-894X(99)00165-1; Drag-Zalesinska, M.; Kulbacka, J.; Saczko, J.; Wysocka, T.; Zabel, M.; Surowiak, P.; Drag, M. Esters of betulin and betulinic acid with amino acids have improved water solubility and are selectively cytotoxic toward cancer cells. Bioorg. Med. Chem. Lett. 2009, 19, 4814–7. https://doi.org/10.1016/j.bmcl.2009.06.046.].
Reviewer#1
- “the contribution of the enhanced content of BA in the formulation for its cytotoxic effect is clearly visible” - therefore, what role does the cation play in these systems?
Authors’ response:
We are thankful to the reviewer for this question. Given that the cell membrane of cancer cells at pH 7, is highly negatively charged, the contribution of the cation in the IL composition to the initial attachment to the cell membrane and subsequent entry into the cell must be considered. A new text has been added in Line 423 explaining the above consideration.
Reviewer#1
- “Since several studies have highlighted the relationship between IL toxicity and the degree of IL-cation lipophilicity, this suggests membrane damage as a potential cause of cytotoxicity” - so is there a greater risk that such compounds may cause more harm to normal cells than bring benefits from treatment?
Authors’ response: We are thankful to the reviewer for this question. In fact, when we tested non-cancerous cells (MCF-10A), especially with the most cytotoxic IL formulation ([LysOEt][BA]2) we did not find any cell specificity of action of the IL. The IC50 for MCF-10A cells treated with ([LysOEt][BA]2) did not differ significantly to those found for MCF-7. So, such compounds should be used with caution in anti-tumor therapy. New paragraph in Material and Methods part is added to introduce MCF-10A cell line (Line 199 and Line 279) and also new text in Results and Discussion from Line 497.
Our data are in consistent with other investigations. For instance IL formulations containing a 1-butyl-3-methyl imidazolium cation and a non-polar amino acid anion have been shown to have almost the same toxicity to colon adenocarcinoma cells (CaCo-2) and mouse fibroblasts (NIH/3T3) [10.1039/C4TX00079J] or cancer (MCF-7) and non cancer (MCF-10A) cells treated with derivative SH‑479 of BA [10.3892/ol.2021.12866]. There is also literature data on the unexpectedly higher toxicity of amino acid-based deep eutectic solvents compared to conventional choline chloride-based DESs [10.1016/j.molliq.2022.121044].

Reviewer 2 Report
Comments and Suggestions for Authors
The manuscript addresses an important area of design and development of novel ILs, along with the application to drug delivery and sensitization of targets. Overall the data is convincing and promising. There are a number of suggestions to improve the scope and presentation of the paper:
1. The authors should strongly consider expanding the introduction. While the discussion of IL and BA are a great introduction, there is significant reports in the literature focused on the interaction of ILs with existing therapeutics, both for cancer and otherwise. This would strongly enhance the manuscript. Several starting points for references: doi.org/10.2217/nnm-2020-0340; 10.1021/acsbiomaterials.0c00447;doi.org/10.1016/S0040-4039(98)02070-X ; doi: 10.1016/j.isci.2020.101853
2. The NMR and IR data is too small to be reasonably interpreted. While the fine-grained details are appropriately included in the supplement, for the main figures I suggest (A) including labels of each sample identity along the right-axis, (B) highlighting 1-2 specifically critical peaks/positions for the set and labelling that on the graphs, and (C) including a statement in the figure legends that the full spectra and interpretation can be found in supplement.
3. For the DSC experiments in figure 7, there are several concerns/questions:(A) the figure legend should indicate the concentration of IL/BA used in the experiments (the methods section says IC50, but it is important to include here) (B) the methods section appears to indicate the same number of VIABLE cells were used in each sample. If so, that should be stated in the text or the legend.
4. The water solubility analysis in table 1 needs some level of comparison/standardization. The methods section indicates this may have been performed solely using visualization of precipitate formation? This seems very subjective and, if that's the case, needs to be clearly described. I also recommend using a different nomenclature than the boxes currently used in Table 1.
5. the cytotoxicity relationship to water solubility, and the underlying hydrophobicity of the molecule, is a phenomenon well known in the antimicrobial world. Antimicrobial polymers and peptides have thoroughly studied the link between increased hydrophobicity and increased cytotoxicity. These references could lend support to your argument. I would suggest starting with work from the Kuroda lab (10.1021/acs.biomac.8b01256) as place to begin finding references on this topic.
Author Response
Dear Reviewer,
Thank you for your critical marks and comments. We amended the manuscript taking into account all reviewers' recommendations. All changes in the text are highlighted.
Please find below our detailed answers.
Review 2
The manuscript addresses an important area of design and development of novel ILs, along with the application to drug delivery and sensitization of targets. Overall the data is convincing and promising. There are a number of suggestions to improve the scope and presentation of the paper:
Dear Reviewer #2,
Thank you for your comments and suggestions on our manuscript entitled "Tuning of the anti-breast cancer activity of betulinic acid via its conversion to ionic liquids" (ID: pharmaceutics-2919721). These comments are all valuable and very helpful in revising and improving our paper, as well as providing important guidance for us. We have taken all the comments into account and made corrections that we hope will meet with approval. Revised sections are highlighted in the text. The following is a point-by-point response to the comments of Reviewer #2.
Reviewer #2
- The authors should strongly consider expanding the introduction. While the discussion of IL and BA are a great introduction, there is significant reports in the literature focused on the interaction of ILs with existing therapeutics, both for cancer and otherwise. This would strongly enhance the manuscript. Several starting points for references: doi.org/10.2217/nnm-2020-0340; 10.1021/acsbiomaterials.0c00447; doi.org/10.1016/S0040-4039(98)02070-X; doi: 10.1016/j.isci.2020.101853
Authors' response: Following your recommendation, we have expanded the Introduction. We have included in the introduction some reviews on the biomedical application of ILs. We are aware of recent advances in the application of ILs in drug delivery, API or natural product-based ILs, and progress in the preparation of biologically active ILs. Nevertheless, we have tried to focus the introduction on betulinic acid, its anticancer properties and what is known about its IL formulations.
Reviewer #2
- The NMR and IR data is too small to be reasonably interpreted. While the fine-grained details are appropriately included in the supplement, for the main figures I suggest (A) including labels of each sample identity along the right-axis, (B) highlighting 1-2 specifically critical peaks/positions for the set and labelling that on the graphs, and (C) including a statement in the figure legends that the full spectra and interpretation can be found in supplement.
Authors' response: Thank you for your comment. Unfortunately the software cannot add the names of compounds on the y-scale. After adding additional signals to the graphs, the spectrum becomes unreadable. Therefore, no further lists have been added. It has been added to the text of the main manuscript that the full spectra and interpretation can be found in the Supplement.
Reviewer #2
- For the DSC experiments in figure 7, there are several concerns/questions:(A) the figure legend should indicate the concentration of IL/BA used in the experiments (the methods section says IC50, but it is important to include here) (B) the methods section appears to indicate the same number of VIABLE cells were used in each sample. If so, that should be stated in the text or the legend.
Authors' response:
We are grateful for this remark, as placing the exact designations of the concentrations of the compounds used will bring further clarity to the reader. The exact values of the concentrations of the substances were added to the legend of Fig. 7 and also were indicated in the text of Material and methods (Line 330).
Reviewer #2
- The water solubility analysis in table 1 needs some level of comparison/standardization. The methods section indicates this may have been performed solely using visualization of precipitate formation? This seems very subjective and, if that's the case, needs to be clearly described. I also recommend using a different nomenclature than the boxes currently used in Table 1.
Authors' response: Of course, we agree that it would be more advantageous to use standardised solubility test methods. However, we used this particular method because it has been used by other authors who have described betulin derivatives [doi.: 10.1016/S0960-894X(99)00165-1; doi.:10.1016/j.bmcl.2009.06.046.]. Regarding the accusation of subjectivity, we also tested the intensity of the solution to confirm the precipitation of the solution. Following the reviewer's comment, we have clarified the description of the methodology.
- the cytotoxicity relationship to water solubility, and the underlying hydrophobicity of the molecule, is a phenomenon well known in the antimicrobial world. Antimicrobial polymers and peptides have thoroughly studied the link between increased hydrophobicity and increased cytotoxicity. These references could lend support to your argument. I would suggest starting with work from the Kuroda lab (10.1021/acs.biomac.8b01256) as place to begin finding references on this topic.
Authors’ response: Thank you for your comment and the recommended paper. We agree that there should be a relationship between the solubility of compounds, hydrophobicity and their biological activity. For example, this effect is clearly observed and discussed by Edwards et al. in a study evaluating the activity and toxicity of a series of antimicrobial peptides against gram-positive, gram-negative and fungal pathogens. The authors reported that the observed trends in antimicrobial activity correlated with the amphipathicity of the compounds and, to a lesser extent, with the overall hydrophobicity. They observed an increase in antibacterial activity with amphipathicity, but also an increase in overall toxicity [doi.: 10.1021/acsinfecdis.6b00045].
In the case of ILs, the effect of the cation on anticancer activity has been discussed in numerous papers. The most studied cations are the bulky aromatic, traalkylamonium and alkyl substituted phorphonium cations. The assessment of the effect is complex and should take into account characteristics such as hydrophobicity, lipophylicity, size, hydration, potential for H-bonding, etc. [doi.: 10.1016/j.jcis.2020.08.115; doi.:10.1080/1062936X.2013.821092; doi.: 10.1016/j.jhazmat.2018.01.028]. In addition, not only cell toxicity, but also the assembly of ILs or their interactions with a specific target (protein, receptor, etc.) strongly depends on the IL concentration and medium [doi.:10.1515/pac-2023-0203, doi.:10.1021/cr500411q; doi.: 10.1021/acs.jpcb.5b04854; doi.: 10.1007/s10930-022-10058-5].
In our study the influence of cation was explained in the text starting in Line 480: „It is well-known fact that the majority of eukaryotic cells exhibit negatively charged cell membrane at physiological pH [Benga G, Holmes RP (1984) Interactions between components in biological membranes and their implications for membrane function. Prog Biophys Mol Biol 43:195–257]. In this regard, the contribution of the cation (from the IL formulation) to the initial adsorption of [LysOEt][BA]2 to the negatively charged cancer cell membrane via electrostatic interactions and its further permeation should also be considered [Dobrzyńska I, Skrzydlewska E, Figaszewski ZA. Changes in electric properties of human breast cancer cells. J Membr Biol. 2013 Feb;246(2):161-6. doi: 10.1007/s00232-012-9516-5. Epub 2012 Nov 8. PMID: 23135059; PMCID: PMC3555248].”

Reviewer 3 Report
Comments and Suggestions for Authors
The work submitted for review concerns the conversion of betulinic acid (BA) into ionic liquids (IL) as a strategy to improve its pharmacokinetic properties and toxicity towards hormone-dependent MCF-7 breast cancer cells. The authors described in great detail the processes of synthesis and analysis of the manufactured products using modern analytical methods. It is worth familiarizing the scientific community with the results of this work, especially since one of the produced ionic liquids [LysOEt][BA]2 turned out to be active against MCF-7. The authors determined the mechanism of cytotoxic action, paying attention to changes in the organization of membrane lipids and modulation of cytoplasmic properties.
I found no objections to the analytical part of the work. However, the work draws attention to the lack of comparative studies in relation to physiological cells and comparison of activity with a known standard. Instead, the authors compare the results obtained for synthesized ionic liquids.
The quite high plagiarism percentage of 22%, according to my analysis, does not indicate plagiarism, because most of it concerns methodology. The work is original.
Regarding editing, I suggest: Table 1.-please remove symbols of Water solubility. Instead please write categories of solubility using words: weak; middle; good’; very good.
Author Response
Dear Reviewer,
Thank you for your critical marks and recommendations. We revised the manuscript following all reviewers' suggestions. All changes in the text were highlighted.
Please find below our responses to your questions:
Reviewers#3
The work submitted for review concerns the conversion of betulinic acid (BA) into ionic liquids (IL) as a strategy to improve its pharmacokinetic properties and toxicity towards hormone-dependent MCF-7 breast cancer cells. The authors described in great detail the processes of synthesis and analysis of the manufactured products using modern analytical methods. It is worth familiarizing the scientific community with the results of this work, especially since one of the produced ionic liquids [LysOEt][BA]2 turned out to be active against MCF-7. The authors determined the mechanism of cytotoxic action, paying attention to changes in the organization of membrane lipids and modulation of cytoplasmic properties.
I found no objections to the analytical part of the work. However, the work draws attention to the lack of comparative studies in relation to physiological cells and comparison of activity with a known standard. Instead, the authors compare the results obtained for synthesized ionic liquids.
Authors’ reponse: We are grateful for this observation, as the addition of non-cancerous cells to this study will bring more clarity to potential therapeutic practice. In fact, when we tested non-cancerous cells (MCF-10A), especially with the most cytotoxic IL formulation ([LysOEt][BA]2) we did not find any cell specificity of action of the IL. The IC50 for MCF-10A cells treated with ([LysOEt][BA]2) did not differ significantly to those found for MCF-7. So, such compounds should be used with caution in anti-tumor therapy. New paragraph in Material and Methods part is added to introduce MCF-10A cell line (Line 199 and Line 279) and also new text in Results and Discussion from Line 497.
Our data are in consistent with other investigations. For instance IL formulations containing a 1-butyl-3-methyl imidazolium cation and a non-polar amino acid anion have been shown to have almost the same toxicity to colon adenocarcinoma cells (CaCo-2) and mouse fibroblasts (NIH/3T3) [10.1039/C4TX00079J] or cancer (MCF-7) and non cancer (MCF-10A) cells treated with derivative SH‑479 of BA [10.3892/ol.2021.12866]. There is also literature data on the unexpectedly higher toxicity of amino acid-based deep eutectic solvents compared to conventional choline chloride-based DESs [10.1016/j.molliq.2022.121044].
Reviewer#3
The quite high plagiarism percentage of 22%, according to my analysis, does not indicate plagiarism, because most of it concerns methodology. The work is original.
Regarding editing, I suggest: Table 1.-please remove symbols of Water solubility. Instead please write categories of solubility using words: weak; middle; good’; very good.
Authors’ response:We thank the Reviewer for the attention given to the manuscript and his/her accurate comments. As recommended, we have corrected Table 1.

Round 2
Reviewer 1 Report
Comments and Suggestions for Authors
I thoroughly analyzed all corrections and improvements implemented by authors. I fully accept the new version, which is significantly better than initial submission. Therefore, the manuscript has been sufficiently improved to warrant publication in Pharmaceutics
Reviewer 3 Report
Comments and Suggestions for Authors
Congratulations to the authors. I have no objections.